# $\epsilon$-Invariant Hierarchical Reinforcement Learning for Building Generalizable Policy

## Abstract

Goal-conditioned Hierarchical Reinforcement Learning (HRL) has shown remarkable potential for solving complex control tasks. However, existing methods struggle in tasks that require generalization since the learned subgoals are highly task-specific and therefore hardly reusable. In this paper, we propose a novel HRL framework called *ε-Invariant HRL* that uses abstract, task-agnostic subgoals reusable across tasks, resulting in a more generalizable policy. Although such subgoals are reusable, a transition mismatch problem caused by the inevitable incorrect value evaluation of subgoals can lead to non-stationary learning and even collapse. We mitigate this mismatch problem by training the high-level policy to be adaptable to the stochasticity manually injected into the low-level policy. As a result, our framework can leverage reusable subgoals to constitute a hierarchical policy that can effectively generalize to unseen new tasks. Theoretical analysis and experimental results in continuous control navigation tasks and challenging zero-shot generalization tasks show that our approach significantly outperforms state-of-the-art methods.

## 1 Introduction

Goal-conditioned Hierarchical Reinforcement Learning (HRL) methods have shown remarkable potential to solve complex tasks(Nachum et al., 2018b)(Kim et al., 2021)(Costales et al., 2021), such as continuous control tasks that require long-horizon navigation (Li et al., 2021)(Gürtler et al., 2021). Goal-conditioned HRL uses subgoals to decompose the original task into several sub-tasks, training a high-level policy to output subgoals and a low-level policy that executes raw actions conditioned on the subgoals. Therefore, the performance of goal-conditioned HRL mainly relies on a well-designed subgoal space. In particular, in many complex realistic scenarios such as robotics, reusable subgoals are necessary for building a generalizable policy that is widely applicable to different tasks.

Prevailing strategies usually utilize task-specific subgoals such as representations extracted from essential states or trajectories (Nachum et al., 2018a)(Li et al., 2021)(Jiang et al., 2019b), or select a subgoal space based on prior knowledge such as (a subspace of) the raw state space (Nachum et al., 2018b) (Zhang et al., 2020). Such subgoals depend on the specific task and thus often cannot be reused in different tasks. For instance, in a maze navigation task, a subgoal representing coordinates in a maze may not be reachable in another maze, although the two mazes can be similar. As a result, policies based on these subgoals cannot be reused in different tasks.

To construct reusable subgoals, a readily applicable choice is to use invariable abstract physical quantities as subgoals such as directions in a navigation task. These abstract subgoals are usually task-agnostic and thereby naturally reusable. However, how to build a generalizable policy based on these abstract subgoals remains a challenge due to the *transition mismatch* problem, which is common in HRL (Zhang et al., 2022) (Levy et al., 2018) and can lead to the non-stationary learning process. In general, the transition mismatch problem emerges when the high-level policy evaluates the subgoals with incorrect transition and rewards due to an inadequately trained low-level policy. To introduce the problem clearly, we will show it in a prevailing challenging task considered by prior HRL methods, i.e., a long-horizon maze navigation problem based on a legged robot, despite our idea is not limited to this particular task. Consider that the high-level produces a subgoal to go to position with coordinates $(x, y)$, but the immature low-level policy reached $(x', y')$. Then, the high-level will evaluate the subgoal by the incorrect reward from $(x', y')$, which will lead to

inaccurate value estimation and even the collapse of the hierarchical policy learning process (Igl et al., 2020b)(Wang et al., 2021). Previous work proposes to alleviate the mismatch problem in single-task settings by refining the wrong transition, such as relabeling the wrong transition by the correct transition obtained from the replay buffer (Levy et al., 2018) (Nachum et al., 2018b) (Kim et al., 2021). However, these methods fail to solve the transition mismatch problem in the multi-task setting, where this problem can be exacerbated by the confusion of changeable tasks with the same abstract subgoals. As successful trajectories are not identical across different tasks, relabeling by sampled trajectories may aggravate the mismatch problem.

In this paper, we propose a novel HRL framework called $\epsilon$-Invariant HRL to leverage task-agnostic abstract subgoals as well as alleviate the influence of the transition mismatch problem. We propose a method called $\epsilon$-Invariant Randomization to inject controllable stochasticity into the low-level policy during the training of the high-level policy. The key idea is to see the mismatch as a kind of randomness and train the high-level policy to be adaptable to this randomness. We term the subgoals introduced with randomness $\epsilon$-Invariant subgoals (the definition is in 3.1). Based on this framework, we propose a parallel synchronous algorithm from A2C Mnih et al. (2016) to evaluate the online policy and update the network by an expected gradient from multiple trajectories sampled with the same parameter instead of updating by a single gradient, which can alleviate the mismatch problem with the expected transition, reducing of influence of incorrect transitions and rewards. The parallel algorithm is proven to possess the potential to solve changeable scenarios and randomness in the environments (Hou et al., 2022)(Espeholt et al., 2018). After solving the mismatch problem, our HRL policy can leverage general task-agnostic subgoals and generalize even to unseen tasks.

To demonstrate the superiority of our HRL framework, we extend the widely-used benchmark based on (Duan et al., 2016)'s work using the MuJoCo (Todorov et al., 2012) simulator. These experiments are designed to control high-dimensional robots to solve long-horizon maze navigation tasks in different mazes with sparse rewards (see details in section 4.1). Some tasks are in a stochastic environment with changeable structures to check the robustness of policy. In these difficult tasks, our method achieves state-of-the-art results compared with the most advanced RL and HRL methods. Our method also shows novel abilities in generalization tasks, in which the structures of the mazes are unseen, even with unseen pre-trained robots. To the best of our knowledge, such complex generalization tasks can hardly be solved by previous methods, and we are the first to build an HRL policy that exhibits generalization capability across different mazes and different robots.

In summary, our contributions are three-fold:

1. We devise an HRL framework for generalizing in high-dimensional controlling maze navigation tasks with a theoretical guarantee.

2. We propose a randomization method for the transition mismatch problem in generalization tasks along with an algorithm that enables stable learning.

3. We provide a new benchmark for evaluating the approaches for high-dimensional maze navigation tasks, and our method outperforms SOTA algorithms. To the best of our knowledge, we are the first to build policies that can generalize to such zero-shot navigation tasks with unseen mazes and unseen robots.

## 2 PRELIMINARIES

We formulate the task in this paper as a goal-conditioned Markov decision process (MDP) (Sutton & Barto, 2018), defined as a tuple $< \mathcal{S}, \mathcal{G}, \mathcal{A}, P, R, \gamma >$. $\mathcal{S}$ is the state space, $\mathcal{A}$ is the action space, and $\mathcal{G}$ is the goal space, which is a set of consistent invariant actions. In this paper we focus on maze-navigating tasks, so that we choose the relative displacement of directions "up, down, left, right" (or "$x+, x-, y+, y-$" in the coordinate system of the environment) as the goals (see details in Section 3.1). $P$ is the transition probability matrix and $P(s'|s,a)$ is the one-step transition probability. $R(s,a)$ is the reward function, $\gamma \in [0, 1)$ is the discount factor.

**Goal-conditional HRL**. We consider the framework that consists of two hierarchies: a high level policy $\pi^H = \pi(g|s; \theta_h)$ and a low-level policy $\pi^L = \pi(a|s, g; \theta_l)$, where $\theta_h, \theta_l$ is the parameter of the two policies parameterized by neural networks respectively. At a high-level timestep $t$, the high-level policy generates a high-level action, i.e. subgoal $g_t \in \mathcal{G}$ by $g_t \sim \pi(g|s_t; \theta_h)$. The

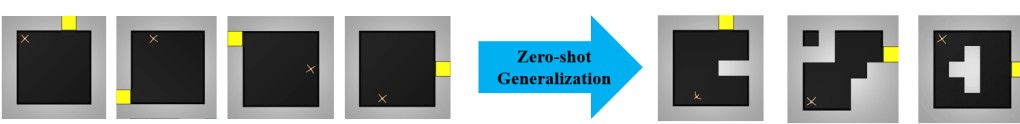

Figure 1: Illustration of Zero-shot Generalization Maze Task. From left to right is Maze-g2, Maze-g3, Maze-g1. Details are in section 4.1

high-level policy gives a goal every $k$ steps and the low-level policy executes the subgoal $g_t$ in $k$ steps. Then the high-level policy receives the accumulative external rewards from the environment $r_t^h = \sum_{i=tk}^{tk+k-1} R(s_i, a_i)$ in $k$ steps. The goal of the high-level is to maximize the expected return $\mathbb{E}[\sum_{t=0}^{H} \gamma^t r_t^h]$ by driving the low-level policy.

The low-level policy receives additional intrinsic reward $r^l$ for efficient learning as follows:

$$r^l(s_{tk}, g_t, a_{tk}, s_{tk+1}) = \alpha \cdot cos < \mathbf{g_t}, \varphi(s_{tk+1}) - \varphi(s_{tk}) > \cdot \|\varphi(s_{tk+1}) - \varphi(s_{tk})\| \qquad (1)$$

where $g_t = \mathbf{g_t}$ is an invariable direction vector of "$\{x+, x-, y+, y-\}$", the $\varphi$ is a coordinate extracting function form state $s$ and $\alpha$ is a constant coefficient. The reward means that the further the agent goes towards the goal direction, the more reward can it obtain. Different from current HRL methods as done in (Kim et al., 2021)(Nachum et al., 2018b), we do not use the relative distance of specific coordinates as the intrinsic reward function but use the fixed distance towards invariable and orthogonal goal directions. By this reward function, the agent will learn to walk towards an abstracted direction according to the subgoal instead of towards a specific location, which preserves the generalization potential for different tasks.

## 3  APPROACH WITH ANALYSIS

**Motivation**. We hope to build an HRL policy to solve maze-navigation and generalization tasks of high-dimensional continuous controlling robots. The difficulty of these tasks is usually caused by large exploration space and sparse rewards. Thus we aim to build a subgoal setting, which can not only reduce the difficulty of exploration but also are reusable in different tasks. Consider that many previous RL works research on tabular mazes, such as the BabyAI platform (Chevalier-Boisvert et al., 2018), if the maze with continuous state space can be discretized into several blocks, the learning process will be more efficient than the original one. So that we set the subgoal space by abstracted task-agnostic subgoals like directions of "front, back, left, right". These subgoals mean the agent should move in the direction of a fixed step. In such setting, the maze is divided into blocks ac-

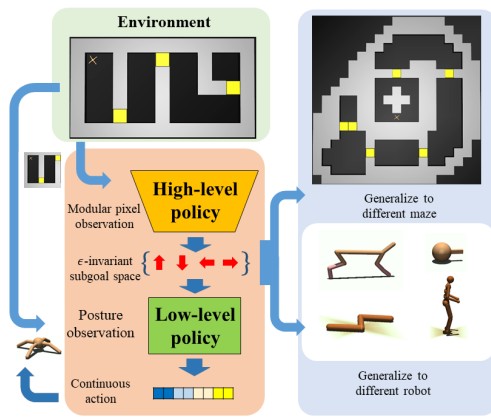

Figure 2: HRL framework illustration. By the $\epsilon$-invariant subgoals, the high-level policy can adapt to different maze tasks and different pre-trained low-level robots.

cording to the distance of the movement in the high-level perspective. The high-level policy can make a decision on finite discrete space, which will significantly improve the exploration efficiency. Meanwhile, such subgoals are physically invariable so that can be reused in any maze. Policy based on these subgoals can also be general among tasks.

**Challenges**. Although the motivation is concise, it needs to overcome three challenges for implementation. (1) Will the hierarchical policy in our setting seriously break the performance of the original optimal policy? How to evaluate the error? Is the error controllable? (2) In HRL the fixed oracle subgoals setting with respective learning process often lead to collapse due to the mismatch of the two hierarchies . How to let the agent learn stably and safely and overcome the mismatch?

In the following paragraphs, we will answer the two questions respectively by theoretical analysis for discussing the effectiveness of our method, and framework building with algorithm design for stable policy learning.

## 3.1 THEORETICAL ANALYSIS

Limited by the length of the paper, in this section we just show the main idea of proof. The details can be seen in Appendix A.

Our main idea to prove the effectiveness of our method is to prove the error between optimal policy and ours in a metric is bounded and can be controlled so that our methods can be used as a sub-optimal policy to approximate the optimal one. The results in stochastic MDPs show that the error between our hierarchical policy and the optimal policy can be bounded.

Firstly, we give the mathematical definition of the $\epsilon$-invariant subgoals (as the high-level abstracted actions).

**Definition 3.1.** *($\epsilon$-invariant subgoals) For every environment $\mathcal{E}$ and an $\epsilon$-invariant subgoal $g_\Delta \in G_\Delta$, transitions of the subgoal from 's' to 's'' and 's'' to 's'''' with optimal low-level policy, for any s, satisfying:*

$$\mathbb{E}_{s'}[\varphi(s) - \varphi(s')] = \mathbb{E}_{s''}[\mathbb{E}_{s'}[\varphi(s')] - \varphi(s'')] \tag{2}$$

*where $\sigma_{\varphi(s'),x}, \sigma_{\varphi(s'),y} \leq \epsilon$*

Here $\pi^H(g|s)$ is the high-level policy. $s'$ and $s''$ are the achieved states by the $\epsilon$-invariant subgoal $g_\Delta$ with optimal low-level policy. $\varphi$ is the coordinate extracting function and $\varphi(s)$ is the coordinate vector of observation $s$. $\sigma_{\varphi(s'),x}, \sigma_{\varphi(s'),y}$ are the variance of achieved state in $x, y$ directions. We consider our high-level policy learning in stochastic MDP so that the $s'$ is a random variable depending on $s$ and $g_\Delta$, and we limit the variance of $s'$ by a little constant $\epsilon$. The equation means such subgoals will give an excepted invariable direction in the coordinate system with a little randomness.

**Error Bound in Stochastic MDP**. As shown in Fig 3, the trajectories of RL method, traditional HRL method, and our method are different, and for that, we build our high-level policy in stochastic MDP with stochastic subgoals.

We consider a goal-conditional HRL where the high-level policy makes a decision every $k$ steps. To compare with the HRL method, the optimal value function of original RL methods in deterministic-MDP can be rewritten with $k$ step as:

$$V_k^*(s_t) = \mathbb{E}_{\tau_k}[R(\tau_k) + \gamma^k V(S_{t+k})] \tag{3}$$

where $\tau_k$ is a trajectory in $k$ steps with optimal policy $\pi^*(\tau_k|s_t)$, $R(\tau_k)$ is the accumulative discounted reward in $k$ steps. And equation 3 can be rewritten as: $V_k^*(s_t) = \sum_{\tau_k} P^\pi(\tau_k|s_t)[R(\tau_k) + \gamma^k V(S_{t+k})]$.

With equation 3, we can define the error between traditional RL method and our method: $|V_k^*(s_t) - \hat{V}_H^\epsilon(s_t)|$. To analyze and control the error, we introduce the value $V_H^*(s_t)$ in optimal stochastic HRL with coordinates as subgoals:

$$|V_k^*(s_t) - \hat{V}_H^\epsilon(s_t)| \leq \underbrace{|V_k^*(s_t) - V_H^*(s_t)|}_{HRL\ error} + \underbrace{|V_H^*(s_t) - \hat{V}_H^\epsilon(s_t)|}_{subgoal\ error} \tag{4}$$

Here we only give the conclusion of our analysis, that is:

$$||V_k^* - V_H^*||_\infty \leq \frac{\nu_k k R_{max}}{2(1-\gamma^k)} + \frac{\gamma^k \nu_k R_{max}}{2(1-\gamma^k)^2} \tag{5}$$

and

$$||V_H^* - \hat{V}_H^\epsilon||_\infty \leq \frac{L_\varphi(2\delta_{max} + 3\sqrt{2}\epsilon)R_{max}}{2(1-\gamma^k)}(k + \frac{\gamma^k}{1-\gamma^k}) \tag{6}$$

where $\delta_{max} = \max_{g,g_\Delta}\{||g||, ||g_\Delta||\}$, $R_{max}$ is the bound of reward function, $\nu_k$ is the transition mismatch rate between optimal RL method and HRL method with stochastic coordinate subgoals. $L_\varphi$ is the Lipschitz constant. The proof is shown in Appendix A.

The result shows that the error between our method and optimal policy indeed can be bounded and controlled. That means our policy can be used as a sub-optimal policy to approximate the original optimal one.

## 3.2 Framework

In this section, we will show how to build a hierarchical framework to handle the abstracted $\epsilon$-invariant subgoals. Generally, the high-level policy decides the direction as subgoals by pixel observation, and the low-level policy receives the expected direction and walks towards the direction for a distance. Thanks to the abstractness of the subgoals defined in 3.1, although the high-level and the low-level policy can learn together, we train them respectively to accelerate the learning process. So that our method decomposes the learning process into two stages with discrete general subgoals, where the high level learns by external rewards and the low level learns by intrinsic rewards. Essentially, the discrete general subgoals are invariable displacement of fixed directions, which will reduce the difficulty of exploration and improve learning efficiency by discretizing the state space into little blocks. Meanwhile, they are reusable and general in any maze-navigation tasks, so as to preserve the generalization abilities of the high-level policy.

**Modular Pixel Observation**. As we utilize the direction as abstracted subgoals for the high-level policy, the influence of the subgoals can hardly be perceived by the agent with the posture of the robot. We add a pixel observation at the top view to look down at the agent. (See in figure 2) It will observe a region of fixed size by a camera following the robot. From this perspective, the influence of the subgoal of directions is invariable and consistent among maze environments. The agent can observe the change in the environment close to the robot. Meanwhile, pixel observation is more adaptable than raw data of posture, which can improve generalization abilities. We provide the pixel observation for the high-level policy to decide which direction to go.

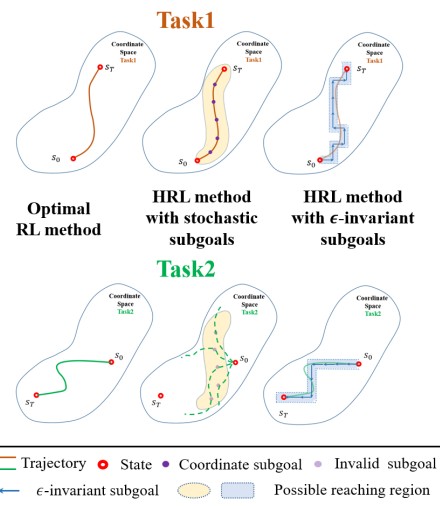

Figure 3: Comparison of different frameworks. $\epsilon$-invariant subgoals (invariable directions with little randomness) can be reused in different navigating tasks.

**$\epsilon$-Invariant Randomization**. As we train the two hierarchies respectively with abstracted subgoals, we train the high-level by the movement directly from the simulator instead of the real walking of the robot for faster learning. However, abstracted subgoals will bring the mismatch problem between the two levels. Because to control a robot moving by legs strictly towards a direction is very hard for RL methods. So that even the well-learned walk policy will contain a slight deviation in the vertical direction of the goal direction. Such inevitable deviation (a performance of mismatch) will cause incorrect evaluation of the subgoals, which will lead to non-stationarity and even collapse in a changeable environment.

To alleviate the problem, we introduce the $\epsilon$-Invariant Randomization method to build the high-level learning process as a stochastic MDP. When training the high-level policy, the simulator moves the robot with random postures and random positions. The postures are sampled from the walking postures of well-learned low-level policy with random subgoals and recorded as offline data. The random position means that a random deviation will be added to the original movement. For instance, the high-level policy gives a "x+" direction as subgoals, then the simulator will move the robot to "x+" with a little offset of $\Delta_x, \Delta_y \sim \mathcal{N}(0, \sigma)$, where $\sigma \le \epsilon$ is much less than the moving distance. As a result, if the high-level policy can overcome the randomness, the wrong execution of the low-level policy will be seen as a sample of the distribution of randomness, which will improve the affordance of the high-level policy.

Network structure and more details can be seen in Appendix C.

**Algorithm**. As we introduce randomness into the learning process to alleviate the mismatching problem, the high level should overcome the stochasticity in the environment to obtain a well-performed policy. To solve the randomness, we propose a parallel algorithm called the parallel expected gradient advantage actor-critic (PEG-A2C) algorithm because the parallel algorithm is proven to possess the potential to adapt to a changeable environment (Hou et al., 2022)(Espeholt

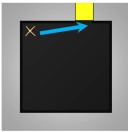 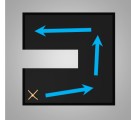 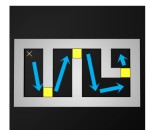 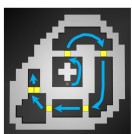

Figure 4: An illustration of the shape of maze environments of our benchmark. They are Random Square, ⊃ Maze, S-shaped Maze, and Spiral Maze respectively. The blue arrow is the successful trajectory.

et al., 2018). The main idea of our algorithm is to sample multiple trajectories with the same parameter by several agents, then concentrate the trajectories and calculate the expected gradients of different trajectories and different steps and update the parameters synchronously. That is, different from previous works which update parameters $\theta$ by $\theta \leftarrow \theta + \alpha\nabla_\theta$, we utilize $\theta \leftarrow \theta + \alpha\mathbb{E}_\tau[\nabla_\theta(\tau)]$ to train the high-level policy. Such gradients can hopefully reduce the impact of randomness, as taking the expectation can be seen as a special kind of automatic gradient clipping by tasks (Zhang et al., 2019)(Bjorck et al., 2021). That is our algorithm evaluates subgoals by expected return and high-level transition. It will alleviate the non-stationarity of the learning process.

The pseudo code is shown in Appendix B.

## 4 EXPERIMENTS

In our experiments we aim to answer the following questions: (1) Can our method learn stably in different complex maze navigation by a robot with high-dimensional action space? (2) How does our method perform compared with advanced RL and HRL algorithms? (3) Can our method generalize to different unseen maze tasks even unseen robots without retraining?

### 4.1 ENVIRONMENTS SETTING

We evaluate a suite of MuJoCo (Todorov et al., 2012) maze-navigating tasks modified from the benchmark from (Duan et al., 2016). Different from the setting of (Kim et al., 2021)(Nachum et al., 2018b), we utilize **sparse** external rewards to make them more challenging. In these tasks, the agent should control an ant robot to achieve a specified area in the maze which is far away from the initial position. To compare completely, we also add experiments on **dense** reward, where the reward is $1/(1 + d)$ of Euclidean distance $d$ between the agent and the current goal in the coordinate system for every step. The mazes are **Ant ⊃-Maze**, **Ant Random Square Maze**, **Ant S-shaped Maze**, **Ant Spiral Maze**, **Ant Spiral Maze**, and **Generalization Maze** (See in Fig 4). They are all difficult mazes with the long navigating horizon. In these mazes, the goal is to pass the door or go to a specific region. Especially **Ant Random Square Maze** is a task in a stochastic environment with random initial positions of both the agent and the door. **Generalization Maze** is a task with three unseen mazes of different structures. **Ant S-shaped Maze** and **Ant Spiral Maze** are extremely long-horizon mazes that require at least thousands of steps by the optimal policy. More details can be seen in Appendix D.

**Implementation**. We build our HRL agent by two policies. The high-level policy is learned by our PAG-A2C algorithm, and the low-level policy is a goal-conditioned policy modified from DroQ (Hiraoka et al., 2021) algorithm, of which the subgoals are directions of 'x+','y+','x-','y-'. As the simulator in MoJoCo allows direct movement of coordinates, we train the two policies respectively in two stages. The high-level policy learns with direct movement with randomness, and the low-level learns by intrinsic reward defined in equation 1. As the subgoals and the low-level policy can be reused in different tasks, we mainly show the high-level training curves and success rates in different tasks and use the same well-learned low-level policy for the ant robot.

### 4.2 BASELINES

We compare our method to several state-of-the-art (SOTA) model-free RL and HRL algorithms in the high-dimensional continuous control tasks said above.

**DroQ**. It is the SOTA RL algorithm for high-dimensional continuous control tasks (Hiraoka et al., 2021), which is the most efficient RL method. It is effective for both simple robots like Hopper and complex robots with large numbers of degrees of freedom like Humanoid.

Figure 5: Comparative experiment results with strong baselines in sparse-reward tasks. The mean and variance are calculated by 3 runs.

**HIGL**. It is the SOTA HRL algorithm for high-dimensional maze-navigating tasks with both sparse rewards and dense rewards for MuJoCo suite (Kim et al., 2021).

**HESS**. It is one of the most advanced subgoal learning HRL methods for continuous control tasks (Li et al., 2021).

**RAND-H**. It is a variety of our HRL method, of which the high-level policy is a random policy and the low-level policy is well-learned. The baseline is used as an ablation study of our method to show the capabilities of our subgoal setting.

**our-oracle**. It is a variety of our HRL method, of which the high-level policy is trained without the low-level policy. The robot will walk by the oracle movement of coordinates executed directly by the simulator instead of the low-level policy. The baseline is also used as an ablation study of our method to show the efficiency of the high-level policy.

## 4.3 RESULTS OF COMPARATIVE EXPERIMENTS

For a fair comparison, we utilize the evaluated reward curves with both the high-level and the low-level, although our hierarchical policies can be trained respectively. The curves are shown in figure 5. We can see that our method outperforms both the SOTA RL and HRL methods. The curves are cut by the same episode to align the results of different methods. All the curves are smoothed by a sliding window. More details can be seen in Appendix D.

**Robustness to Stochasticity and Mismatch**. In these tasks, the 'Ant Random Square' is a square maze with a random initial position of robots and goals. In this task, the mismatch problem will become critical as the successful trajectory is episodically changeable. So that correction by historical samples is invalid. As shown in the curve in figure 5, the HIGL method can obtain a reward at first, but gradually reduced it with a downward trend. Our method can learn with a gradually increasing return. The curve shows that our method can adapt to such a stochastic environment and learn stably, proving that our algorithm indeed can overcome randomness. But other methods can hardly adapt to such a stochastic environment.

**Comparative Result**. The non-hierarchical method DroQ performs poorly in all the tasks with sparse rewards, demonstrating the strength of the hierarchical structures in solving long-horizon tasks with sparse rewards. HESS method also performs not so well with little accumulate return, due to the requirement of large exploration episode steps for sparse reward, which is consistent with the results reported in their paper (Li et al., 2021). The SOTA HIGL method outperforms other baselines while underperforming our method, which learns with a gradually rising average reward but slower than ours, showing the superiority of our methods.

In tasks with dense rewards (figure 6), we utilize average step reward and goal-reaching success rate to evaluate the performance of these methods. The performance of previous methods is improved a lot but the curves of success rate are still under our method.

**Ablation Study**. The RAND-H and our-oracle show the ablation result of our method. The high-level policy with oracle movement (our-oracle) can learn stably with our algorithm. We can see that the low-level policy (RAND-H) can obtain rewards in a certain frequency with our subgoal setting even with random high-level policy, so as to preserve stable feedback for the learning process. The two ablation experiment curves show the reason why our method learns faster.

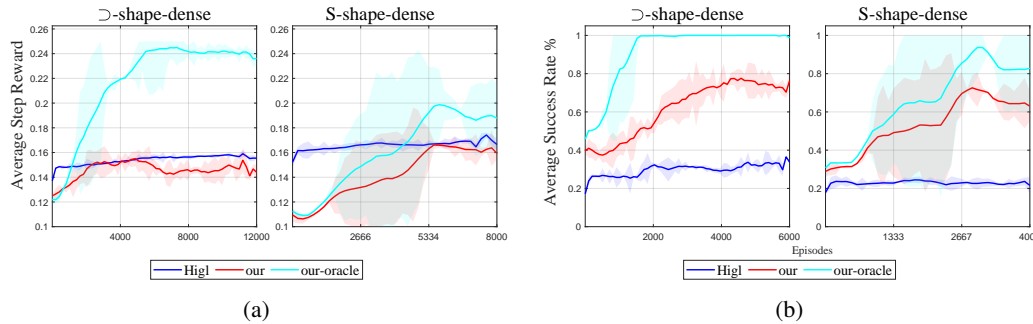

Figure 6: Comparative experiment results in dense-reward tasks. (a) is curve of average reward of every step. (b) is curve of average success rate $= achieved\ goals/total\ goals$

Table 1: Generalization Task Result for Different Mazes (Zero-shot Success Rate %). 'Maze-ori' is the trained maze, and the other three are zero-shot generalization maze .

| Method/Maze | Maze-ori | Maze-g1 | Maze-g2 | Maze-g3 |
|---|---|---|---|---|
| Ours | $51.8 \pm 4.4$ | $45.6 \pm 5.2$ | $27.8 \pm 6.7$ | $73.6 \pm 4.6$ |
| Ours-Oracle | $66.4 \pm 5.4$ | $72.0 \pm 6.3$ | $58.4 \pm 8.3$ | $84.8 \pm 6.6$ |
| HIGL | $25.4 \pm 5.3$ | $\leq 1$ | $\leq 1$ | $\leq 1$ |

## 4.4 RESULTS OF ZERO-SHOT GENERALIZATION EXPERIMENTS

**Zero-shot Maze**. In these experiments, we utilize the policy trained in 'Ant Square Maze' and test in new unseen mazes of fixed shapes without retraining. We compare our method with the HIGL method by the well-performed policy of training process in the Ant Random Square environment with sparse reward. Result is shown in table 1. The test mazes are fixed shapes without randomness. so that sometimes the test result can be better than the training result.

**Zero-shot Robot**. In this experiment, we test the single high-level policy with the oracle movement and the whole HRL policy respectively. In table 2 we show zero-shot result of the mere high-level policy in 'Maze $\supset$ shape'. It is to show the adaptive capabilities of our high-level policy with our subgoal setting.

From the results, we can see that such generalization tasks can hardly be solved by previous RL and HRL methods due to the specific subgoal or goal setting. They can neither adapt to different maze tasks nor change the low-level policy without retraining the high-level policy. Our HRL method uses invariable abstracted subgoals and can adapt to different scenarios. Our method is also flexible and can adapt to different unseen robots with the low-level policy receiving the same subgoals with a certain success rate.

**Visualization Results**. To show the generalization ability of our HRL policy, we show the heatmap of trajectories of our method and HIGL (figure 7). We can see that the HIGL agent mainly moves near the initial position and is blocked by unseen structures, but our agent can achieve the door more frequently.

Table 2: Generalization Task Result for Different Robot (Zero-shot Success Rate %). Except 'Ant', the other robot are unseen. Average success rate $= achieved\ goals/total\ goals$

| Method/Robot | Ant | Point | Swimmer | Humanoid |
|---|---|---|---|---|
| Ours-Oracle | $99.6 \pm 0.5$ | $53.2 \pm 1.4$ | $47.0 \pm 2.4$ | $48.2 \pm 1.3$ |
| HIGL | $67.0 \pm 5.2$ | — | — | — |

## 5 RELATED WORK

**Goal-conditioned HRL methods.** In the recent few years, hierarchical reinforcement learning methods have been widely studied for complex high-dimensional control and long-horizon tasks. There are many goal-conditioned HRL methods for subgoals learning, handling and discovery for these difficult tasks (Nachum et al., 2018b)(Li et al., 2021)(Kim et al., 2021)(Gürtler et al., 2021)(Li et al., 2020)(Zhang et al., 2020). These methods usually leverage task-specific subgoals such as spe-

**Our method**

HIGL

Figure 7: Visualization result of zero-shot maze generalization tasks. The red and blue points represent the achieved positions in the maze. The more lightly the position is, the more frequently the agent achieving. The histogram is to straightly show the frequency of the achievement of every position.

cific coordinate (Nachum et al., 2018b), sampled abstracted or original states (Nachum et al., 2018a) or abstracted trajectory descriptions (Jiang et al., 2019a) to complete complex tasks. Different from previous methods, our framework utilizes the $\epsilon$-invariant subgoals (direction with randomness) as abstracted high-level actions, which do not stand for task-specific descriptions but the invariable relative effects of action sequences. Such subgoals can be reused ignoring the change of states in different tasks, which are more general.

**Generalizable Policy Learning.** Constructing policies can be reuse (or generalize) in new (even unseen) tasks is a long-standing challenge, which is researched by many works in different perspectives (Igl et al., 2020a)(Xu et al., 2022)(Wang et al., 2019)(Wang et al., 2020). Recent works are mainly three lines for generalizable policy learning. (1) Learning reusable or transferable skills, option or policies for different tasks(Shah et al., 2021) (Nam et al., 2021)(Klissarov & Precup, 2021). (2) Learning policies adapting to a distribution of environments or tasks with different dynamics by risk-sensitive objective functions(Lyle et al., 2022) (Lei & Ying, 2020)(Kirsch et al., 2019). (3) Learning to extract reusable abstracted representations like language, logical symbols or graph from visual observation or state-action trajectories (Jain et al., 2020)(Agarwal et al., 2020)(Vaezipoor et al., 2021).

Our method is related to the third line, i.e., utilize the abstracted representations for improving the generalization abilities of the policy. But the difference is that we do not leverage representations expressing tasks in a distribution or invariable abstracted trajectories. Instead, we use the abstracted representation to represent the physically invariable relative locomotion. Such representations are usually general in the kind of maze-navigation tasks, thus leading to strong generalization abilities.

## 6 CONCLUSION

In this paper, we propose a novel HRL framework to build a generalizable hierarchical policy with abstract task-agnostic subgoals. We give theoretical analysis to prove the effectiveness of our method and design algorithm to overcome the transition mismatch problem in generalization tasks to construct a stable policy learning process. Strong results in challenging experiments show the superiority of our method. Meanwhile, our method can achieve zero-shot generalization in different unseen tasks, which cannot be dealt with by previous methods. We believe that idea of our method could be used beyond our task setting. For future work, we will focus on more general policy learning to solve more complex tasks, not limiting to HRL.

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

# A    THEORETICAL ANALYSIS

With equation 3, we can define the error between traditional RL method and our method: $|V_k^*(s_t) - \hat{V}_H^\epsilon(s_t)|$. To analyze and control the error, we introduce the value $V_H^*(s_t)$ in optimal stochastic HRL with coordinates as subgoals:

$$|V_k^*(s_t) - \hat{V}_H^\epsilon(s_t)| \leq \underbrace{|V_k^*(s_t) - V_H^*(s_t)|}_{HRL\ error} + \underbrace{|V_H^*(s_t) - \hat{V}_H^\epsilon(s_t)|}_{subgoal\ error} \tag{7}$$

so that we can control the error by two parts: (1) the error from RL policy to stochastic HRL policy and (2) the error from ordinates subgoals to $\epsilon$-invariant subgoals.

**HRL Error**. For the former (denote as $e_H = |V_k^*(s_t) - V_H^*(s_t)|$), the error bound can be controlled by the corollary of conclusion of Zhang's work (Zhang et al., 2022). We firstly formalize a critical factor termed `transition mismatch rate` between RL policy and stochastic HRL policy. (The difference between our work and Zhang's is that the factor in our work represents the difference between RL and HRL policy, but the difference between two HRL methods in Zhang's work.)

**Definition A.1.** *(k-step transition mismatch)*

*For a goal-conditioned MDP $\mathcal{M}$ with $P^k(s_{t+k}|s_t, g_t)$ as its k-step transition probability under an optimal goal-conditioned policy $\pi^H(g_t|s_t)$, the k-step transition mismatch rate of $\mathcal{M}$ is defined as:*

$$\nu_k \triangleq \max_{s_t, g_t, s_{t+k}} |P^\pi(\tau_k|s_t) - \pi^H(g_t|s_t)P^k(s_{t+k}|s_t, g_t)| \tag{8}$$

The mismatch rate $\nu_k$ captures the difference between RL policy and HRL policy by the part of trajectories in $k$ steps. The less uncertainty the high-level policy performs with, the less $\nu_k$ is. The deterministic-MDP of high-level policy will lead to $\nu_k = 0$. Then for error $e_H$, we have the conclusion as follows:

**Lemma A.2.** *(Corollary of Theorem 2 in (Zhang et al., 2022)) With discounted factor $\gamma$, bounded reward function $R_{max} = \max_{s,a} R(s, a)$ and $\nu_k$ defined in equation 8, there is*

$$||V_k^* - V_H^*||_\infty \leq \frac{\nu_k k R_{max}}{2(1 - \gamma^k)} + \frac{\gamma^k \nu_k R_{max}}{2(1 - \gamma^k)^2} \tag{9}$$

*Proof.* See Appendix.A.1 $\qquad\qquad\square$

Lemma A.2 means the error between RL method and stochastic HRL method will be controlled by $\nu_k$. If the distribution of the randomness of subgoals is in a little region with little variance, the HRL policy can approximate the optimal RL policy with controllable error.

**Subgoal Error**. To analyses the error caused by our subgoals (the latter error is denoted as $e_S = |V_H^*(s_t) - \hat{V}_H^\epsilon(s_t)|$), we should introduce a new metric. We first consider the actually reached region caused by traditional coordinate subgoals and our $\epsilon$-invariant subgoals. If the region and trajectories of the two HRL policies are close to each other in the coordinate system, to some extent, that means the policy and transition are similar. On the other hand, to measure the influence of the traditional subgoals and the $\epsilon$-invariant subgoals, the distance and trajectories in the coordinate system can directly reflect the difference between them. Thus, we assume that:

**Assumption A.3.** *(Lipschitz condition) In the k-step reachable region under an optimal goal-conditioned policy, for a traditional goal-conditioned MDP $\mathcal{M}$ with subgoal $g_t$ and $P^k(s_{t+k}|s_t, g_t)$ as its k-step transition probability, as well as an $\mathcal{M}_\epsilon$ with $\epsilon$-invariant subgoals $g_\Delta$ and transitions $P^k(s_{t+k}|s_t, g_\Delta)$, there is:*

$$|P^k(s_{t+k}|s_t, g_t) - P^k(s'_{t+k}|s_t, g_\Delta)| \leq L_\varphi ||\varphi(s_{t+k}) - \varphi(s'_{t+k})|| \tag{10}$$

*where $L_\varphi$ is the Lipschitz constant, $\varphi$ is the coordinate extracting function defined above.*

The Lipschitz condition is common and widely used in many other RL or HRL works (Wang et al., 2019)(Shah et al., 2020)(Gogianu et al., 2021). Assumption A.3 means that, if the low-level policy learns well and can execute the subgoals given by the high-level policy, the difference between the

two HRL policies can be measured by the Euclidean distance of displacement in the coordinate system. The distance is in the reachable region with optimal low-level policy in $k$ steps, from the same start state $s_t$. If the reached states are closed, the policies are considered similar.

By the setting above, we have the following theorem which provides a suboptimality upper bound for error $e_S$:

**Theorem A.4.** *With discounted factor $\gamma$, bounded reward function $R_{max} = \max_{s,a} R(s,a)$, Lipschitz constant $L_\varphi$, variance bound $\epsilon$ defined above, with a high probability there is*

$$||V_H^* - \hat{V}_H^\epsilon||_\infty \leq \frac{L_\varphi(2\delta_{max} + 3\sqrt{2}\epsilon)R_{max}}{2(1 - \gamma^k)}(k + \frac{\gamma^k}{1 - \gamma^k}) \tag{11}$$

*where $\delta_{max} = \max\limits_{g,g_\Delta}\{||g||, ||g_\Delta||\}$. $||g||$ and $||g_\Delta||$ are the distance of the relative displacement from any $s_t$ to $s_{t+k}$ by subgoal $g_t$ and $g_\Delta$ respectively.*

*Proof.* See in Appendix. A.2 □

The theorem A.4 means that the error between the traditional HRL method with coordinates as subgoals and our method can be constrained by the similarity metric of the two policies. By the conclusion above, our method has the theoretical guarantee to approximat to the original optimal policy with a controllable error bound. Meanwhile, in our methods, the subgoals can be reused in different maze environments, which leads to superiority in generalization tasks.

## A.1 PROOF OF LEMMA 1

*Proof.* Consider a state $s_t \in \mathcal{S}$, for the value function of $k$ step:

$$V_k^*(s_t) = \sum_{\tau_k} P^\pi(\tau_k|s_t)[R(\tau_k|s_t) + \gamma^k V^*(S_{t+k})] \tag{12}$$

where $P^\pi(\tau_k|s_t) = P^\pi(s_{t+k}|s_t)$ is the $k$-step transition probability with policy $\pi$, $R(\tau_k|s_t) = R(s_{t+k}, s_t) = \sum_{i=t+1, s_i \in \tau_k}^{t+k-1} \gamma^i R(s_i)$ is the bounded accumulative return in $\tau_k$, $\tau_k$ is the possible $k$-step trajectory by optimal policy $\pi^*$ executing $k$ times. The optimal policy should maximize the external reward from the environment:

$$\pi^* = \arg\max_\pi \mathbb{E}_{(s,a)\sim\pi}[\sum_{t=1}^T \gamma^t R(s_t, a_t)] \tag{13}$$

equation 12 can be rewritten as vector form for any state $s_t \in \mathcal{S}$, i.e.:

$$V_k^*(s_t) = \left\langle \sum_{\tau_k} P^\pi(s_t), R(s_t) + \gamma^k V^* \right\rangle \tag{14}$$

where $\langle \cdot, \cdot \rangle$ is the inner product of vectors, $P^\pi(s_t) \in [0,1]^{|\mathcal{S}|}, \forall s \in \mathcal{S}, ||P^\pi(s_t)||_1 = 1$ denote the probablistic distribution vector of the $k$-step trajectory starting from $s_t$ under the policy $\pi^*$. $R(s_t) \in [0, kR_{max}]^{|\mathcal{S}|}$ is the accumulative rewards vector of $k$-step of the $k$-step trajectory. $V^*$ is the optimal value vector function of all the states.

Similarly, value of HRL method with stochastic subgoals can be written as follow:

$$V_H^* = \sum_{g \in \mathcal{G}} \pi^H(g_t|s_t) \sum_{s_{t+k} \in \mathcal{S}} P^k(s_{t+k}|s_t, g_t)[R(s_{t+k}, s_t) + \gamma^k V_H^*(S_{t+k})]$$
$$= \left\langle \sum_{g \in \mathcal{G}} \pi^H(g_t|s_t)P^k(g_t, s_t), R(s_t) + \gamma^k V_H^* \right\rangle \tag{15}$$

where $\pi^H(g_t|s_t)$ is the high-level policy to choose subgoal $g_t$, $P^k(s_{t+k}|s_t, g_t)$ is the $k$-step transition probability of the low-level policy with subgoal $g_t$. $R(s_{t+k}, s_t)$ is also the $k$-step accumulative rewards. $P^k(s_t, g_t)$ is the probablistic distribution vector of the $k$-step trajectory starting from $s_t$ with subgoal $g_t$.

Denote that $\bar{P}_H(s_t, g_t) \triangleq \sum_{g \in \mathcal{G}} \pi^H(g_t|s_t) P^k(g_t, s_t)$, so that definition A.1 can be rewritten as:

$$\nu_k = \max_{s_t, g_t} ||P^\pi(s_t) - \bar{P}_H(s_t, g_t)||_\infty \tag{16}$$

Then for every $s_t \in \mathcal{S}$, there is:

$$
\begin{aligned}
&|V^*(s_t) - V_H^*(s_t)| \\
&= \left| \left\langle \sum_{\tau_k} P^\pi(s_t), R(s_t) + \gamma^k V^* \right\rangle - \left\langle \sum_{g \in \mathcal{G}} \pi^H(g_t|s_t) P^k(g_t, s_t), R(s_t) + \gamma^k V_H^* \right\rangle \right| \\
&\leq \left| \left\langle \sum_{\tau_k} P^\pi(s_t) - \sum_{g \in \mathcal{G}} \pi^H(g_t|s_t) P^k(g_t, s_t), R(s_t) \right\rangle \right| \\
&+ \gamma^k \left| \left\langle \sum_{\tau_k} P^\pi(s_t), V^* \right\rangle - \left\langle \sum_{g \in \mathcal{G}} \pi^H(g_t|s_t) P^k(g_t, s_t), V_H^* \right\rangle \right|
\end{aligned}
\tag{17}
$$

note that $\left\langle \sum_{\tau_k} P^\pi(s_t), \mathbf{1} \right\rangle = 1$ and $\left\langle \sum_{g \in \mathcal{G}} \pi^H(g_t|s_t) P^k(g_t, s_t), \mathbf{1} \right\rangle = 1$, where $\mathbf{1}$ is an all-one vector of $|\mathcal{S}|$-dimension, for the first term of equation 17, we have

$$
\begin{aligned}
&\left| \left\langle \sum_{\tau_k} P^\pi(s_t) - \sum_{g \in \mathcal{G}} \pi^H(g_t|s_t) P^k(g_t, s_t), R(s_t) \right\rangle \right| \\
&= \left| \left\langle \sum_{\tau_k} P^\pi(s_t) - \sum_{g \in \mathcal{G}} \pi^H(g_t|s_t) P^k(g_t, s_t), R(s_t) - \frac{kR_{max}}{2} \cdot \mathbf{1} \right\rangle \right|
\end{aligned}
\tag{18}
$$

by Holder inequality, there is:

$$
\begin{aligned}
(18) &\leq \left\| \sum_{\tau_k} P^\pi(s_t) - \sum_{g \in \mathcal{G}} \pi^H(g_t|s_t) P^k(g_t, s_t) \right\|_1 \cdot \left\| R(s_t) - \frac{kR_{max}}{2} \cdot \mathbf{1} \right\|_\infty \\
&\leq \max_{s_t, g_t} \left\| P^\pi(s_t) - \bar{P}_H(s_t, g_t) \right\|_\infty \left\| R(s_t) - \frac{kR_{max}}{2} \cdot \mathbf{1} \right\|_\infty \\
&= \frac{\nu_k k R_{max}}{2}
\end{aligned}
\tag{19}
$$

The second term can be similarly bounded by:

$$
\begin{aligned}
&\gamma^k \left| \left\langle \sum_{\tau_k} P^\pi(s_t), V^* \right\rangle - \left\langle \sum_{g \in \mathcal{G}} \pi^H(g_t|s_t) P^k(g_t, s_t), V_H^* \right\rangle \right| \\
&\leq \gamma^k \left| \left\langle \sum_{\tau_k} P^\pi(s_t), V^* \right\rangle - \left\langle \sum_{\tau_k} P^\pi(s_t), V_H^* \right\rangle \right| \\
&+ \gamma^k \left| \left\langle \sum_{\tau_k} P^\pi(s_t), V_H^* \right\rangle - \left\langle \sum_{g \in \mathcal{G}} \pi^H(g_t|s_t) P^k(g_t, s_t), V_H^* \right\rangle \right|
\end{aligned}
\tag{20}
$$

where

$$\gamma^k \left| \left\langle \sum_{\tau_k} P^\pi(s_t), V^* \right\rangle - \left\langle \sum_{\tau_k} P^\pi(s_t), V_H^* \right\rangle \right| \leq \gamma^k \left\| V^* - V_H^* \right\|_\infty \tag{21}$$

and

$$
\begin{aligned}
&\gamma^k \left| \left\langle \sum_{\tau_k} P^\pi(s_t), V_H^* \right\rangle - \left\langle \sum_{g \in \mathcal{G}} \pi^H(g_t|s_t) P^k(g_t, s_t), V_H^* \right\rangle \right| \\
&\leq \gamma^k \left\| \sum_{\tau_k} P^\pi(s_t) - \sum_{g \in \mathcal{G}} \pi^H(g_t|s_t) P^k(g_t, s_t) \right\|_1 \cdot \left\| V_H^* - \frac{kR_{max}}{2(1-\gamma^k)} \cdot \mathbf{1} \right\|_\infty \\
&\leq \gamma^k \max_{s_t, g_t} \left\| P^\pi(s_t) - \bar{P}_H(s_t, g_t) \right\|_\infty \left\| V_H^* - \frac{kR_{max}}{2(1-\gamma^k)} \cdot \mathbf{1} \right\|_\infty \\
&\leq \gamma^k \nu_k \frac{kR_{max}}{2(1-\gamma^k)}
\end{aligned}
\tag{22}
$$

So that with 19, 21 and 22, there is

$$|V^*(s_t) - V_H^*(s_t)| \leq \frac{\nu_k k R_{max}}{2} + \frac{\gamma^k \nu_k k R_{max}}{2(1-\gamma^k)} + \gamma^k \left\| V^* - V_H^* \right\|_\infty \tag{23}$$

Since 24 holds for all $s \in \mathcal{S}$, so there is:

$$\left\| V^* - V_H^* \right\|_\infty \leq \frac{\nu_k k R_{max}}{2(1-\gamma^k)} + \frac{\gamma^k \nu_k k R_{max}}{2(1-\gamma^k)^2} \tag{24}$$

$\square$

### A.2 Proof of Theorem 1

*Proof.* Consider assumption A.3 there is:

$$
\begin{aligned}
&\left\| \varphi(s_{t+k}) - \varphi(s'_{t+k}) \right\| \\
&= \left\| \varphi(s_{t+k}) - \varphi(s_t) + \varphi(s_t) - \varphi(s'_{t+k}) \right\|
\end{aligned}
\tag{25}
$$

restate that the $\varphi(s_t)$ is the coordinate of state $s_t$, so that $\varphi(s_{t+k}) - \varphi(s_t)$ is the direction vector of the movement from step $t$ to $t+k$.

By definition 3.1, we denote the relative displacement $\Delta_t = ||g_\Delta|| = \mathbb{E}_{s_{t+k}}[\varphi(s_t) - \varphi(s_{t+k})]$ for any state $s_t$ with $\epsilon$-invariant subgoal $g_\Delta$. So that the displacement vector can be written as $\vec{g}_{\Delta t} = \vec{\Delta}_t + \vec{\Delta}_\epsilon$, where $\vec{\Delta}_t$ is a fixed vector of expected direction and $\vec{\Delta}_\epsilon$ is a random vector. $\Delta_{\epsilon,x}, \Delta_{\epsilon,y} \sim \mathcal{N}(0, \sigma)$, where $\sigma \leq \epsilon$ is much less than the moving distance $\Delta_t$. Thus, the equation 25 can be rewritten as:

$$
\begin{aligned}
25 &= \left\| \varphi(s_{t+k}) - \varphi(s_t) - (\vec{\Delta}_t + \vec{\Delta}_\epsilon) \right\| \\
&\leq \left\| \varphi(s_{t+k}) - \varphi(s_t) - \vec{\Delta}_t \right\| + \left\| \vec{\Delta}_\epsilon \right\|
\end{aligned}
\tag{26}
$$

Consider that in normal distribution, the random variable has a high probability fall into the region of $[-3\sigma, 3\sigma]$. So that with a high probability, there is $||\vec{\Delta}_\epsilon|| \leq \sqrt{2 \times (3\sigma)^2} \leq 3\sqrt{2}\epsilon$. Thus, the assumption A.3 can be rewritten as follows:

$$|P^k(s_{t+k}|s_t, g_t) - P^k(s'_{t+k}|s_t, g_\Delta)| \leq L_\varphi(2\delta_{max} + 3\sqrt{2}\epsilon) \tag{27}$$

where $\delta_{max} = \max_{g, g_\Delta}\{||g||, \Delta_t\}$.

The equation 27 can be seen as the HRL transition mismatch rate between our method and optimal HRL method with stochastic coordinate subgoals. It is bounded by the maximal $k$-step movement of the agent with the variance of the random variable. With lemma A.2, the similar conclusion can be obtained by changing the $k$-step transition mismatch to equation 27. Then we can get the result of A.4.

$\square$

## B   ALGORITHM

Our algorithm for the high-level policy learning is as follows (algorithm 1):

---
**Algorithm 1** PEG-A2C Algorithm
---
1: Initialize multi-process actor parameters $\theta_a^i$ for $i \in [1, n]$
2: Initialize multi-process value parameters $\theta_v^i$ for $i \in [1, n]$
3: **for** episodes in 1,M **do**
4:   **for** $i \in [1, n]$ **do**
5:    Reset gradients: $d\theta_a^i$ and $d\theta_v^i$
6:    Synchronize thread-specific parameters
7:    **repeat**
8:     Perform $a_t$ according to policy $\pi(a_t|s_t)$
9:     Receive reward $r_t$ and new state $s_{t+1}$
10:    $t \leftarrow t + 1$
11:   **until** terminal $s_T$ or $t == t_{max}$
12:   Set $R = r$
13:   **for** $j \in \{t - 1, \ldots, 0\}$ **do**
14:    $R \leftarrow r_j + \gamma R$
15:    Accumulate gradients w.r.t. $\theta_a'$

$$d\theta_a \leftarrow d\theta_a + \frac{1}{n}\frac{1}{t}\nabla_{\theta_a'} \log \pi(a_j|s_j; \theta_a')(R_i - V(s_i; \theta_v'))$$

16:    Accumulate gradients w.r.t. $\theta_v'$

$$d\theta_v \leftarrow d\theta_v + \frac{1}{n}\frac{1}{t}\frac{\partial}{\partial \theta_v}(R_j - V(s_j; \theta_v'))^2$$

17:   **end for**
18:  **end for**
19:  Synchronize and update parameters
20: **end for**
---

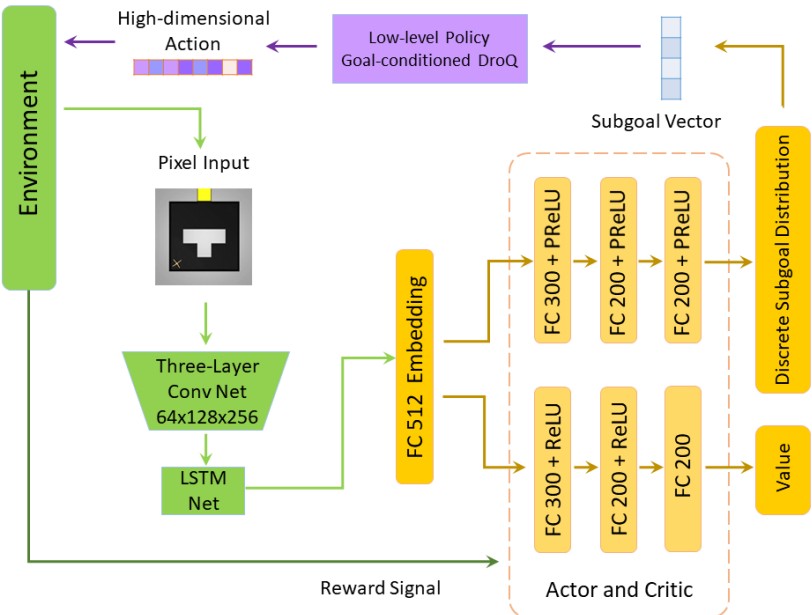

Figure 8: Structure of the network.

## C DETAILS OF FRAMEWORK

### C.1 NETWORKS

The structure of our high-level network is shown as follows (figure 8):

## D DETAILS OF EXPERIMENTS

**Ant ⊃-Maze**. In this task, the agent should navigate from the bottom to the top. Different from the setting in previous works, the maze is larger, and the agent will only attain reward '1' once when it passes the corner and reach the final region.

**Ant Random Square Maze**. It is an empty room with a door and a robot. In this task, the agent will start at a random initial position every episode, to walk towards the yellow door, which also has a random position on the wall. Only when the agent passes the door will it obtain a reward.

**Ant S-shaped Maze**. In this task, the agent will start at the left region and should pass three doors. The trajectory is circuitous and long-horizon, especially the final door is more difficult to achieve than the region in Ant ⊃-Maze. Every first-time transiting of the doors will give the agent a reward.

**Ant Spiral Maze**. This task is in a large maze with spiral routes and five doors and the agent will start at the middle region. Such a long-horizon task requires at least thousands of steps to move to the final region. Also, every first time transiting the doors will give the agent a reward.

**Generalization Maze**. This task includes three fixed mazes of 'Maze-g1', 'Maze-g2', and 'Maze-g3' (figure 1), which are variants of 'Ant Random Square Maze' with different unseen structures. Only when the agent passes the door will it obtain a reward.

### D.1 STEPS OF EVERY EPISODE OF DIFFERENT TASKS

Table 3 shows the maximal steps of every episode of every task.

Table 3: Steps of every episode of different tasks

| **Maze** | Maze-random | Maze-U | Maze-S | Maze-spiral | Maze-g1 | Maze-g2 | Maze-g3 |
|---|---|---|---|---|---|---|---|
| Steps per ep | 600 | 600 | 1000 | 5000 | 600 | 600 | 600 |

## D.2 DETAILS OF REWARD SETTING

For the sparse reward, it will be obtained by the agent when the agent goes across the door, i.e., the coordinates of the agent fall into a region of the door. For the dense reward, it is $1/(1 + d)$ of Euclidean distance $d$ between the agent and the current goal in the coordinate system for every step. Once the agent goes to the current goal and gets the reward, the goal will update, and the reward will be calculated by the new goal. As a result, the curves of average reward in these tasks may decline sometimes.

## D.3 HYPER-PARAMETERS

Table 4: Steps of every episode of different tasks

| **Hyper-parameters** | Value | Details |
|---|---|---|
| Subgoal dimension | 8 | two-hot vector of four subgoals |
| Subgoals | $\uparrow, \downarrow, \leftarrow, \rightarrow$ | abstract subgoals Represented by vector |
| Room size | {7*7, 7*7, 14*7, 21*21} | grids of 'random','⊃','S','spiral' |
| Grid size | 3*3 | size in coordinates system of MuJoCo |
| Sparse reward | 1 | |
| Dense reward | [0.03, 1.0] | range of dense reward |
| High-level frequency | 25 | steps of the low-level to execute |
| Intrinsic reward coefficient $\alpha$ | 3 | |

