# OpenReview forum: "$\epsilon$-Invariant Hierarchical Reinforcement Learning for Building Generalizable Policy"
_ICLR.cc/2023/Conference — Submitted to ICLR 2023_

### Official Review · Reviewer_9r3Q · 2022-10-24

**Confidence:** 4
**Correctness:** 4
**Technical Novelty And Significance:** 2
**Empirical Novelty And Significance:** 2
**Recommendation:** 5

**Clarity, Quality, Novelty And Reproducibility:**

-Clarity of the paper can be improved. The motivation of including section 3.1 in the way it has been included, results unclear/confusing. The paper leaves in the appendices many details that are important for the understanding of the model.

Novelty:
- The idea of learning in a hierarchical way using RL to control a robot and how to get out of a maze has been explored before (e.g. LESSON model of (Li et al., 2021)).

- The theoretical motivation of the proposed approach is a simple modification of the work of Zhang et al. (2022). While there a k-step adjacent region of the current state using an adjacency constraint is used/explored during learning the policies, this manuscript assumes that the distribution of the randomness of subgoals is in a little region with little variance. Therefore, the paper simply substitutes the search over a k-step adjacent region of the current state of the agent using an adjacency constraint, by the following random algorithm: when training the high-level policy, the simulator moves the robot with random postures and random positions. Overall, this is a minor simplification, and a detailed comparison with the work by Zhang et al. (2022) should have been included.

-Moreover, I can hardly find fundamental differences between the theoretical development in Zhang et al. (2022) and the one in this manuscript. At the end of the day, considering a HRL and a standard RL this paper uses the same Lemma as in (Zhang et al., 2022).


Some minor comments:
-Figure 5: yellow (DroQ performance) curve can't be seen.
-Punctuate the equations, they are part of the text.

Reproducibility: No code is provided, and not enough details are provided to reproduce the results.

**Strength And Weaknesses:**

Strengths:
+ The paper described an HRL policy learning process that offers generalization capability across different mazes and different robots. This is an interesting contribution of the work. The zero-shot robot generalization experiment (Section 4.4, table 2) opens an interesting research direction.


Weaknesses:

-The experimental evaluation has some important limitations:

a) Not enough previous HRL methods which have been specifically designed for the same problem are considered in the experimental evaluation. The manuscript just includes the use of HIGL model (Kim et al., 2021)., but others should have been considered: e.g. (Li et al., 2021), HRAC (Zhang et al. 2022) .

b) The evaluation metric proposed is not appropriate. Average reward can be useful for some problems, but in maze navigation tasks, the success rate is probably the most informative metric. See how all previous works use it previous papers (HIGL for instance), which will allow a direct comparison with them. So, I encourage the authors to change figure 5 to include this metric (and remove figure 6). The reward that the proposed approach accumulated is considerably higher that the one gathered by the rest of models, so I do not expect any surprise when the metric changes, however, the reader will be able to see how "fast" the new model is in achieving a SR of 100% with respect to the rest of the methods in the literature.

c) I encourage the authors to also include previously used maze shapes in the experimental evaluation ((Li et al., 2021)(Kim et al., 2021))

d) For the ablation study, the inclusion of RAND-H and our-oracle is an appropriate choice. I am intrigued, however, by the performance of Ours-Oracle in the generalization experiment in Table 1. How can the model trained using the oracle for a particular type of maze work well in any other type of maze? This means that the high-level policy learned by the proposed model (without the oracle), is far from being effective in generalization tasks.

-See my comments below about the novelty, which a one of the weaknesses.

**Summary Of The Paper:**

The manuscript describes the construction of an Hierarchical Reinforcement Learning (HRL) policy for solving the navigation in a maze task.
Technically, the paper proposes the epsilon-Invariant HRL to leverage task-agnostic abstract subgoals. This approach tries to mitigate the transition mismatch problem ((Zhang et al., 2022) (Levy et al., 2018)) in HRL. Basically, this problem emerges when we try to learn reusable (or task agnostic) subgoals (using some invariable abstract physical quantities as subgolas, e.g. navigation directions), to later build a generalizable high-level policy. The latter evaluates the subgoals with incorrect rewards caused by poorly trained low-level policies.
The approach in the paper is simple: consider the transition mismatch as a sort of randomness process, this way the high-level policy is trained to adapt to it.
A simple parallel algorithm (the parallel expected gradient advantage actor-critic (PEG-A2C)) is introduced to calculate the expected gradients of different trajectories and update the parameters synchronously, sampling multiple trajectories.
Maze navigation experiments are performed in the MuJoCo suit for maze-navigation tasks. Multiple baselines are used in the experiments (DroQ - SOTA RL model, HIGL - SOTA HRL model, etc.)


**Summary Of The Review:**

Overall, the paper could be of interest to the rest of the community. The ideas are somewhat incremental. The main problem is in the experimental evaluation, and in the clarity with which the model has been laid out (especially in its comparison with the work of Zhang et al. 2022). For the moment, I am inclined to consider the article below the acceptance threshold.

---

> ### Author Response · Authors · 2022-11-12
> **Response to Reviewer 9r3Q**
>
> Explanation for weakness:
>
> >a) Not enough previous HRL methods which have been specifically designed for the same problem are considered in the experimental evaluation. The manuscript just includes the use of HIGL model (Kim et al., 2021)., but others should have been considered: e.g. (Li et al., 2021), HRAC (Zhang et al. 2022) .
>
> 1、Thank you for your suggestion. We have read the papers on LESSION and HRAC methods. But we do not make a comparison with LESSION and HRAC because our baselines of HESS and HIGL can cover them.
>
> For the work of LESSION, it has achieved some nice results in some maze tasks. The HESS [1] method is actually the following work improving the subgoal representation learning process by the same first author of LESSION. In the paper of HESS, they compare with LESSION method and outperform it in the $\supset$-maze task with sparse reward. Thus, HESS is more efficient and more powerful for these maze tasks. So that we consider that the performance of HESS can be more representative than LESSION.
>
> However, as reported in their paper, HESS requires a large number of training steps to learn the subgoal representation. In our experiment, it learns slower than our method and HIGL method. And LESSION is reported to use more steps, meanwhile even fails in long-horizon complex tasks like ‘Ant FourRooms’ in their paper. Thus we think the comparison with HESS is enough.
>
>
> For HRAC [3] method, the HIGL [4] method is more representative. The HIGL method is indeed based on HRAC and improves the adjacency constraint by landmark correction. They also compare with HRAC in tasks with sparse reward and little stochasticity, and HIGL outperforms HRAC. The reported results show that HIGH is more adaptive than HRAC in maze-navigation tasks. So that we consider that the comparison with HGIL is enough to cover HRAC.
>
>
> To the best of our knowledge, HESS and HIGL are the most advanced HRL methods before the ICLR2023 submission deadline. So that we think the comparison with them is representative and enough, for HIGL is also SOTA HRL method in maze-navigation tasks with sparse reward.
>
>
> [1] Li, Siyuan, et al. "Active Hierarchical Exploration with Stable Subgoal Representation Learning." International Conference on Learning Representations. 2022.
> [2] Kim, Junsu, Younggyo Seo, and Jinwoo Shin. "Landmark-guided subgoal generation in hierarchical reinforcement learning." Advances in Neural Information Processing Systems 34 (2021): 28336-28349.
>
> [3] Zhang, Tianren, et al. "Generating adjacency-constrained subgoals in hierarchical reinforcement learning." Advances in Neural Information Processing Systems 33 (2020): 21579-21590.
>
> >b) The evaluation metric proposed is not appropriate. Average reward can be useful for some problems, but in maze navigation tasks, the success rate is probably the most informative metric. See how all previous works use it previous papers (HIGL for instance), which will allow a direct comparison with them. So, I encourage the authors to change figure 5 to include this metric (and remove figure 6). The reward that the proposed approach accumulated is considerably higher that the one gathered by the rest of models, so I do not expect any surprise when the metric changes, however, the reader will be able to see how "fast" the new model is in achieving a SR of 100% with respect to the rest of the methods in the literature.
>
> 2、Thank you very much for your suggestion. We apologize that we do not emphasize that, in our experiments, the average reward can indeed stand for the success rate. We actually quite agree with your view to utilize success rate to show the capability of policy.
>
> For figure 5, it shows the results of tasks with sparse rewards. Every first time achieving the goal positions or doors will give a reward ‘1’ to the agent. For example, the task of ‘Random-square’ has just one goal, so the average reward being ‘1’ means the success rate is 100%. For the task of ‘S-shape’, there are three doors with a total reward ‘3’, so that ‘average reward/3’ can stand for success rate. So are all the tasks with sparse rewards. We will revise the figure and emphasize the success rate in the camera-ready version.
>
> For figure 6 (a), it shows the average reward in tasks with dense rewards (different task settings from figure 5). It is applicable to our dense reward setting. Because the dense reward is based on the '$1/(1+|dis|)$’, i.e., the closer the distance towards the current goal, the larger the reward is. So that the average reward indirectly reflects the speed of the agent to go toward the goals.
>
> Meanwhile, figure 6 (b) directly show the success rate. It is complementary to figure 6 (a).

---

> > ### Author Response · Authors · 2022-11-12
> > **Response to Reviewer 9r3Q (continued)**
> >
> > >c) I encourage the authors to also include previously used maze shapes in the experimental evaluation ((Li et al., 2021)(Kim et al., 2021))
> >
> > 3、We agree with your view to include previously used maze shapes. So that we use $\supset$-Maze as a comparative experiment to show the performance of our method. Meanwhile, we want to challenge more complex tasks, so we also use ‘S-shape’ and ‘Spiral-shape’. If necessary, we can add other maze shapes in the camera-ready version.
> >
> > >d) For the ablation study, the inclusion of RAND-H and our-oracle is an appropriate choice. I am intrigued, however, by the performance of Ours-Oracle in the generalization experiment in Table 1. How can the model trained using the oracle for a particular type of maze work well in any other type of maze? This means that the high-level policy learned by the proposed model (without the oracle), is far from being effective in generalization tasks.
> >
> > 4、In the ablation study, we trained our high-level policy by oracle movement of the simulator. But that does not mean our high-level policy cannot generalize to different tasks without oracle movement. The oracle movement can be replaced by the locomotion of the low-level policy with a fixed step. And the oracle movement is just to accelerate the learning process. As we train the two levels of policy respectively, the high-level policy can learn with both the oracle movement and the trained low-level. The ablation result of high-level policy with oracle movement is to show the upper bound of the capability of our method because none of RL or HRL policy can move better than the oracle. That does not mean our method relies on the oracle movement. The result of ‘Rand-h’ is the trained low-level policy with a random high-level policy. Without the high-level policy, the low-level policy can also explore and obtain rewards. The frequency of obtaining rewards is higher than the initial policy of baselines. That means our method can learn with more frequent rewards, so that the policy can learn quickly.
> >
> >
> > In fact, the generalization ability comes from the invariable abstract subgoals instead of the oracle movement. The oracle movement can be replaced by the locomotion of the low-level policy with a fixed step. These abstract subgoals are general in different maze navigation tasks. Because the subgoals are four invariable directions of the orthogonal coordinate axis in a 2D plane. That means any trajectories in the maze with a flattened plane can be decomposed into these subgoals. For a simple instance in a grid-like environment, actions of ‘up, down, left, right’ can be enough for all kinds of mazes. So in continuous space, the four directions are also enough for flattened mazes. As the movements are shown in the Modular Pixel Observation, the movement of different directions can be seen in the top view and we do not use coordinates as input. So that our method does not limit to oracle movement or specific coordinates.
> >
> > These explanations can be seen in the ‘Framework’. Maybe our paper is not reading-friendly so it makes it hard to understand. We will revise the paper and emphasize the usage of the oracle movement in the camera-ready version.

---

> > > ### Author Response · Authors · 2022-11-12
> > > **Response to Reviewer 9r3Q (continued)**
> > >
> > > Explanation for the novelty and reproducibility:
> > >
> > > 1、We sincerely apologize that we do not emphasize the code in the supplemental materials. All the details can be seen in the code. We also provide requirements and two trained low-level policies in the ‘model’ folder to help the reader reproduce our method quickly.
> > >
> > > >The theoretical motivation of the proposed approach is a simple modification of the work of Zhang et al. (2022). While there a k-step adjacent region of the current state using an adjacency constraint is used/explored during learning the policies, this manuscript assumes that the distribution of the randomness of subgoals is in a little region with little variance. Therefore, the paper simply substitutes the search over a k-step adjacent region of the current state of the agent using an adjacency constraint, by the following random algorithm: when training the high-level policy, the simulator moves the robot with random postures and random positions. Overall, this is a minor simplification, and a detailed comparison with the work by Zhang et al. (2022) should have been included.
> > >
> > > 2、Thank you for your suggestion for the comparison of theoretical results, we will add it to the paper in the camera-ready version.
> > >
> > > In fact, there are two main differences between our theoretical analysis and Zhang’s work [1].
> > >
> > > Firstly, the definition of mismatch is different from Zhang’s work. They define a kind of mismatch between two stochastic HRL scenes with and without K-step adjacency constraints. But we define another kind of mismatch between optimal RL and stochastic HRL in the ‘HRL error’. Here the k-step is not an adjacency constraint but an HRL common setting to align RL and HRL policy. So that the error bounds are similar but different.
> > >
> > > Secondly, the ‘subgoal error’ is a unique part of our method, which is based on the epsilon-invariant definition. This part is to analyze the difference between the stochastic HRL with coordinates subgoals and epsilon-invariant subgoals of our method.
> > >
> > > We have acknowledged that our theoretical work is inspired by Zhang’s work, but in fact, we give analyses in different scenes and get different results.
> > >
> > >
> > >
> > >
> > >
> > >
> > > [1] Zhang, Tianren, et al. "Adjacency constraint for efficient hierarchical reinforcement learning." IEEE Transactions on Pattern Analysis and Machine Intelligence (2022).

---

### Official Review · Reviewer_LpyT · 2022-10-24

**Confidence:** 3
**Correctness:** 3
**Technical Novelty And Significance:** 3
**Empirical Novelty And Significance:** 3
**Recommendation:** 5

**Clarity, Quality, Novelty And Reproducibility:**

* Besides some grammatical issues and problems with clarity on the sentence-level in some places, the exposition is relatively clear and the contributions/results are easily understood.

* The work is generally high quality, but there are some missing ablations/comparisons that are key to understanding the relative impact of each of the contributions.

* The full method appears to be novel in this setting, but the individual ideas are familiar.

* The paper provides sufficient detail that the results should be possible to reproduce.

**Strength And Weaknesses:**

Strengths:

* Both major contributions (i.e. using more general subgoals for maze navigation tasks and injecting stochasticity into low-level policies to regularize/improve robustness of the high-level policy) are well-motivated and appear to be novel for this problem setting.

* All important details are described clearly.

* The results (especially the generalization results) are impressive and indicate a clear benefit of the proposed approach over baselines in continuous control maze navigation tasks.

* The discussion of related work appears to be complete.

Weaknesses:

* The main weakness is that there are a number of contributions presented in this work, and there is no comprehensive ablation study demonstrating to what degree each contributes to the improved performance. It would be very helpful to see an ablation study where the effects of (1) abstract (and discretized) subgoals, (2) injected stochasticity, and (3) expected gradients are controlled. Without this, it is difficult to make sense of the results, and to understand the relative importance of the contributions. It would also be useful to see how well the baselines perform when these tricks are applied.

* There are a number of grammatical errors throughout the document that make some points unclear. I suggest another careful pass through to make sure each sentence is well-structured for clarity. There are a number of examples throughout the document, but here is just one example (with some rough ideas on how to improve the grammar/clarity in brackets); in the last sentence of ‘motivation’ paragraph: “[Thus, a] policy based on these subgoals [may also generalize better across related] tasks.”

* There are some minor formatting issues. E.g. backticks are not used as opening quotes in some instances (e.g. defn 3.1, motivation paragraph), and when multiple works are cited they are not included in the same set of parentheses (they appear to each be different calls to \cite).

* It is unclear how useful the theoretical analysis is. Instead of bounding the the error between an optimal RL method and the proposed method, it could have been nice to see some theoretical motivation for using stochasticity in the first place. But it is possible I am misinterpreting the results—-I did not carefully read the proofs.

* There does not appear to be any results demonstrating the effects of different levels of injected stochasticity. This would be helpful to see how robust the method is to this hyperparameter. It should also point to an important trade-off.

Questions:

* It is not clear to me why the transition mismatch problem is not mitigated by previous relabeling approaches in the multi-task setting. It is claimed that because “successful trajectories are not identical across different tasks, relabeling sampled trajectories may aggravate the mismatch problem”. Is there evidence of this (I see no citation), or some clearer intuition for why this is the case? An example could also be helpful.

* Which of the implementation tricks are applied to the baseline methods? I.e. between (1) abstract subgoals, (2) expected gradients, and (3) using discretized high-level action space. This is unclear to me from the paper, but I could have missed it somewhere.

* What value of $\sigma$ is used for the results in the paper? It is possible I missed this.

**Summary Of The Paper:**

This paper proposes to address the transition mismatch problem in GCHRL using (1) more readily reusable abstract subgoals, and (2) manually injecting stochasticity into the low-level policy as a form of regularization. Both of these contributions are ultimately proposed to improve generalization on unseen tasks where the same subpolicies and high-level policy are applicable. In addition to these contributions, they also propose multiple additional modifications, including using expected gradient updates, as well as a new form of intrinsic reward to train their abstract subgoals. They provide theoretical analysis showing that the error between their method’s hierarchical policy and the optimal policy can be bounded. Lastly, they compare their method to SOTA hierarchical and non-hierarchical algorithms, and demonstrate that their method outperforms these prior approaches, and achieves significantly better zero-shot generalization performance.

**Summary Of The Review:**

* This work is generally well-written and the method is well-motivated, but there are a number of contributions the authors make in this paper, and without the proper ablations, it is difficult to understand the relative importance of each. For instance, without these ablation results, the performance benefits may be merely due to the form of the abstract subgoals provided / corresponding intrinsic rewards, rather than the injected stochasticity, and this would make for a comparatively weaker contribution. These concerns will need to be addressed in order for me to recommend acceptance of this paper.

---

> ### Author Response · Authors · 2022-11-12
> **Response to Reviewer LpyT**
>
> Explanation for weakness:
>
> >The main weakness is that there are a number of contributions presented in this work, and there is no comprehensive ablation study demonstrating to what degree each contributes to the improved performance. It would be very helpful to see an ablation study where the effects of (1) abstract (and discretized) subgoals, (2) injected stochasticity, and (3) expected gradients are controlled. Without this, it is difficult to make sense of the results, and to understand the relative importance of the contributions. It would also be useful to see how well the baselines perform when these tricks are applied.
>
> 1、We agree with your viewpoint of concluding our contribution in one line. In fact, our contribution is indeed one line, but maybe our paper is not reading-friendly so you do not get it. The main goal of our work is to build a flexible and generalizable HRL policy. The corresponding structure is shown as follows:
>
> To build a generalizable policy, we propose to use abstract and reusable subgoals. -->
>
> To build a policy with abstract subgoals, we design injected stochasticity to overcome the mismatch problem. -->
>
> To solve the brought instability by stochasticity, we utilize expected gradients.
>
> These contributions are in one line of three aspects of the idea, problem, and implementation. They are not conflict or paratactic. We will also revise the paper to emphasize the line of logic more clearly.
>
> >There are a number of grammatical errors throughout the document that make some points unclear. I suggest another careful pass through to make sure each sentence is well-structured for clarity. There are a number of examples throughout the document, but here is just one example (with some rough ideas on how to improve the grammar/clarity in brackets); in the last sentence of ‘motivation’ paragraph: “[Thus, a] policy based on these subgoals [may also generalize better across related] tasks.”
>
> >There are some minor formatting issues. E.g. backticks are not used as opening quotes in some instances (e.g. defn 3.1, motivation paragraph), and when multiple works are cited they are not included in the same set of parentheses (they appear to each be different calls to \cite).
>
> 2、Thank you for your suggestion on how to revise the paper. We are happy to accept your opinion and we will check and revise the paper in the camera-ready version.
>
> >It is unclear how useful the theoretical analysis is. Instead of bounding the the error between an optimal RL method and the proposed method, it could have been nice to see some theoretical motivation for using stochasticity in the first place. But it is possible I am misinterpreting the results—-I did not carefully read the proofs.
>
> 3、The theoretical analysis is to explain and analyze the qualitative theoretical performance of our method compared with the original optimal policy. There are two parts of our bound, and the ‘HRL error’ is about the performance between RL policy and HRL policy with injected stochasticity.
>
> >There does not appear to be any results demonstrating the effects of different levels of injected stochasticity. This would be helpful to see how robust the method is to this hyperparameter. It should also point to an important trade-off.
>
> 4、The effect of different levels of injected stochasticity is important. In our paper, we just give a useful type of injected stochasticity and the experimental results show that little injected stochasticity is helpful for policy learning. The hyperparameter can be seen in the code. As for how to make a trade-off of stochasticity, what other kinds of stochasticity are useful and how to control it is an interesting point. We hope to make research on it in the future work.

---

> > ### Author Response · Authors · 2022-11-12
> > **Response to Reviewer LpyT (continued)**
> >
> > Answer to questions:
> > >It is not clear to me why the transition mismatch problem is not mitigated by previous relabeling approaches in the multi-task setting. It is claimed that because “successful trajectories are not identical across different tasks, relabeling sampled trajectories may aggravate the mismatch problem”. Is there evidence of this (I see no citation), or some clearer intuition for why this is the case? An example could also be helpful.
> >
> > 1、Relabeling is a common method in off-policy HRL to improve learning efficiency. There are mainly two kinds of relabeling methods. The first one is to amend the wrong or poor trajectories by the successful ones. The second one is to revise the transitions of the high-level policy by replacing the expected subgoals with the achieved subgoals.
> >
> > These methods are merely applicable to fixed tasks without stochasticity. The reason is that the relabeling method requires correct trajectories or transitions to refine the experience replay buffer. However, in changeable generalization tasks or tasks with stochasticity, these methods do work not so well and will even aggravate the mismatch problem. Because the sampled trajectories or transitions that are specific to a single task will not always be correct among all the tasks. Some will lead to incorrect policies.
> >
> > For a simple instance, a pickup task with randomness initial position. In a large square room, the agent is initialized in the center. The goal is to pick up an object with different initial positions of the four corners in every episode. The agent should move to approach the object and pick up it. In these scenes, the achieved subgoals of previous methods may be coordinates on one of the successful routes toward a corner between the agent and the goal. The sampled subgoal will naturally lead the agent in the wrong direction when the position of the goal changes. For more complex generalization tasks, the previous methods with relabeling will obtain poor performance.
> >
> > For another example in our generalization experiment, the sampled coordinates in the training tasks are on the successful routes but may be on an intransitive block in the generalization tasks. The previous methods can hardly adapt to the change without retraining.
> >
> > >Which of the implementation tricks are applied to the baseline methods? I.e. between (1) abstract subgoals, (2) expected gradients, and (3) using discretized high-level action space. This is unclear to me from the paper, but I could have missed it somewhere.
> >
> > 2、The type of the subgoal is an important part of the HRL algorithm. The abstract and invariable subgoals and the corresponding expected gradients method are unique components of our method so they are not utilized in baselines. The discretized high-level action space is a common set of HRL methods.
> >
> > >What value of  $\sigma$ is used for the results in the paper? It is possible I missed this
> >
> > 3、The value of $\sigma$ is just bounded by the assumption of $\epsilon$ . It is a qualitative analysis. Actually, we set it one-ninth of the oracle movement by the simulator for implementation. The value of $\sigma$ and how to use it can be seen in the code in the supplemental material.

---

### Official Review · Reviewer_BQSm · 2022-10-30

**Confidence:** 3
**Correctness:** 3
**Technical Novelty And Significance:** 3
**Empirical Novelty And Significance:** 3
**Recommendation:** 6

**Clarity, Quality, Novelty And Reproducibility:**

- The paper is relatively clearly written, and is well-readable.
- The proposed method seems to be techincally sound and the evaluation is done in a reaonable fashion.
- The idea seems novel and unique.

**Strength And Weaknesses:**

Strengths:
- This paper is clearly written and easy to follow. The motivation is clearly indicated and the methodlogy is described clearly.
- The issue of handling mismatch in hierarchical decision settings is a timely and important.
- The proposed framework looks like making sense in general.
- The numerical evaluation is done with reasonable details.
- It is good to provide a theoretical proof on some properties of the proposed scheme.

Weaknesses:
- The theoretical proof relies on the assumption on the epsilon-invariance. It is not clearly explained whether or not this assumption makes practical senses and/or is a simplying one.
- The generalizability of the proposed scheme is not very clearly explained/defended. The notion of subgoals is claimed to be abstract and generic, but it is not very clear to come up with subgoals other than in a geographical domain.

**Summary Of The Paper:**

This paper presents a framework to mitigate a transition mismatch problem in hierarchical reinforcement learning, which utilizes abstract, task-agnostic subgoal generation. Under a certain condition on invariance in subgoals, this work proves thae the error between the hierachical RL and the original full RL can be bounded. The efficacy of the proposed method has been demonstrated on Ant Maze problem in Mujoco, compared against several baselines.

**Summary Of The Review:**

This paper is handling an important problem in hierarchical reinforcement learning by presenting a reasonable framework. The generalizability of the proposed scheme needs to be better justified.

---

> ### Author Response · Authors · 2022-11-12
> **Response to Reviewer BQSm**
>
> Thank you very much for your positive review!
>
>
> Explanation for weakness:
>
> >The theoretical proof relies on the assumption on the epsilon-invariance. It is not clearly explained whether or not this assumption makes practical senses and/or is a simplying one.
>
>
> 1、The assumption on the epsilon-invariance is to alleviate the mismatch problem caused by the generalizable abstract subgoals. Specifically, if the high-level policy gives a subgoal to go towards the ‘x+’ direction but the low-level policy executes incorrectly due to the immature policy or stochasticity of the environment, the evaluation of the subgoal will be wrong. So we introduce stochasticity into the high-level movement. So the high-level policy should learn policy to adapt to stochasticity. The epsilon-invariance assumption is to limit the stochasticity for fear of learning instability. For implementation, we introduce the normal distribution respectively into the movement with a little variance. The details can be seen in the “3.2 Framework: epsilon-invariant randomization”. The implementation details can be seen in the code in the supplemental materials.
>
>
> >The generalizability of the proposed scheme is not very clearly explained/defended. The notion of subgoals is claimed to be abstract and generic, but it is not very clear to come up with subgoals other than in a geographical domain.
>
> 2、How to come up with invariable abstract is an important problem. It is an important direction for our future work. Besides the invariable direction for geographical navigation tasks, there are also many kinds of subgoals. A heuristic view is to utilize object-oriented subgoals as invariable subgoals. For a simple instance, a common task of the Gym-Minigrid or the BabyAI environment is to pick up a key and open the locked door. The ‘key’ and the ‘door’ can be invariable objects and be set as abstract subgoals. The policy or sub-policy can be built on these objects in the environments, such as building a policy to pick up the key ignoring the change in the environment. These policies can be generalizable in different tasks.
>
> Although the subgoals in our paper can be mainly used in the navigation tasks, we believe that the idea of utilizing invariable abstract subgoals to build a generalizable policy can be an inspiring idea for more generalizable RL works.

---

### Official Review · Reviewer_pFzR · 2022-11-02

**Confidence:** 2
**Correctness:** 2
**Technical Novelty And Significance:** 2
**Empirical Novelty And Significance:** 2
**Recommendation:** 3

**Clarity, Quality, Novelty And Reproducibility:**

The paper needs to be more carefully polished. The writing quality is concerned. Moreover, the novelty is also a concern since the main contribution is only the discrete subgoals for better generalization. Because there are many missing details in the experiment section mentioned above, I have no confidence in the reproducibility.

**Strength And Weaknesses:**

+) The studied problem is essential, and the proposed idea is reasonable. Using four directions as subgoals for navigation tasks makes more sense than using absolute coordinates in prior works. The results also suggest that the learnable high-level policy is reusable for different low-level controllers (different agents).


-) The manuscript has not been well written. Several typos, grammar errors, duplicated words, or redundant spaces exist. In addition, some notations are misleading. Taking “x+, x-, y+, y-“ as an example, are the x and y variables? Or do they only indicate direction? All of them make the paper hard to follow.


-) Some algorithms proposed in prior works use collected on-policy rollouts to compute empirically expected policy gradients to perform the policy optimization, such as SAC, PPO, and TRPO. Is there a specific reason for using A2C in the experiments? Since the authors claimed the expected A2C algorithm is one of the contributions, a comparison to the mentioned policy gradient algorithm is necessary.


-) There are many missing details in the experiment section. For example, what is the action space for different agents (e.g., the low-level controllers for different agents?)? What are the learning rates to train the high-level policy and the low-level controller? What optimization algorithm (e.g., SGD?) is used in the training stage?

**Summary Of The Paper:**

This work proposes to use four directions (top, down, left, right) as the general subgoals to learn the high-level policy in a hierarchical reinforcement learning (HRL) setting. Since the directions are task-agnostic and agent-agnostic for the typical navigation jobs, the proposed framework could be more generalizable. Further, to overcome the mismatch problem while training an HRL policy, the authors (1) trained the high-level and low-level policies separately and (2) introduced a slight noise controlled by \epsilon into the perfect low-level controller in the environment. The experimental results in MuJoCo suggest that the proposed framework outperforms various baselines.

**Summary Of The Review:**

The work proposed a simple but intuitive way to discretize subgoal space to improve the generalization ability of an HRL policy. The experimental results show the effectiveness of the proposed framework. However, since the paper is not well polished, it is hard to read and understand the keys under the hood. In addition, because there are many missing details in the experiment section, I have no confidence in the reproducibility. Finally, the usefulness of one of the claimed contributions about “expected policy gradients by A2C” is not validated, which makes the technical contributions weak.

---

> ### Author Response · Authors · 2022-11-12
> **Response to Reviewer pFzR**
>
> Thank you for your feedback and your approval of our idea.
>
> Firstly, we comprehend your worry that our work is hard to follow. We apologize that we do not emphasize where is our code in our paper. **Actually, we put all the code with requirements and a video in the supplementary material.**
>
> The parameters, algorithm, and structure of our network are all in the code so we do not introduce them in detail in our paper. All the package and library functions can be installed by PIP. The benchmark of our maze environments is also in the code. We have reproduced our code in different PCs and servers several times for repeating experiments, and the results are all robust. We also provide two trained models of the low-level policy in the folder ‘model’, so that our algorithm can be easily reproduced quickly. As the two levels of our policy are trained respectively, the parts of our method can be transplanted or inserted into other methods easily. So our method is easy to follow, and Reviewer BQSm also acknowledges this.
>
>
> Explanation for weakness:
>
> >-) The manuscript has not been well written. Several typos, grammar errors, duplicated words, or redundant spaces exist. In addition, some notations are misleading. Taking “x+, x-, y+, y-“ as an example, are the x and y variables? Or do they only indicate direction? All of them make the paper hard to follow.
>
> For your first question about subgoals “x+, x-, y+, y-“, they are four invariable directions of the orthogonal coordinate axis in a 2D plane. They derive from our idea that we aim to use invariable and reusable subgoals to generalize to more tasks. These subgoals of invariable direction can be reused in many different maze-navigation tasks. We also introduce them as well as how to use them to design intrinsic rewards to train the low-level policy in the Section Preliminaries. As for grammar errors, we will check and revise them in the camera-ready version.
>
> > -) Some algorithms proposed in prior works use collected on-policy rollouts to compute empirically expected policy gradients to perform the policy optimization, such as SAC, PPO, and TRPO. Is there a specific reason for using A2C in the experiments? Since the authors claimed the expected A2C algorithm is one of the contributions, a comparison to the mentioned policy gradient algorithm is necessary.
>
>
> For the second weakness, thank you for your suggestion. However, we consider that the comparison is dispensable. The reason is as follows:
> (1) A2C is just one of the choices. Our main contribution to the algorithm is to use the parallel expected gradient to adapt to randomness. This idea is general and not limited to A2C. In fact, PPO, TRPO, and SAC are all improved online actor-critic-like algorithms that evaluate both policy and value function, so our method can theoretically be used in these algorithms. Here we just give an implementation of our idea and empirically validate it through complex experiments.
> (2) For original algorithms without our parallel method, they have been proven hard to adapt to tasks with noise, variable observation, or changeable dynamic environment, let alone generalization tasks [1-5]. So that original RL algorithms like SAC, PPO [2,3,4] without improvement cannot solve such complex generalization tasks can be a consensus of the researchers who focus on generalizable RL.
> (3) As for the comparison with the traditional RL algorithm, we compare our algorithm with the SOTA one DroQ. It is representative of the advanced RL algorithm and has achieved nice results in continuous-controlling tasks, but still fails in complex tasks in our experiment. Because traditional RL methods can neither overcome stochasticity nor solve long-horizon complex tasks with sparse reward. It shows the superiority of our method.
>
> >-) There are many missing details in the experiment section. For example, what is the action space for different agents (e.g., the low-level controllers for different agents?)? What are the learning rates to train the high-level policy and the low-level controller? What optimization algorithm (e.g., SGD?) is used in the training stage?
>
> For the third question, the details are all in the code in the supplementary material. We will also add the implemental details into the appendix for the subsequent follower.

---

> > ### Author Response · Authors · 2022-11-12
> > **Response to Reviewer pFzR (continued)**
> >
> > > The paper needs to be more carefully polished. The writing quality is concerned. Moreover, the novelty is also a concern since the main contribution is only the discrete subgoals for better generalization. Because there are many missing details in the experiment section mentioned above, I have no confidence in the reproducibility.
> >
> > >The work proposed a simple but intuitive way to discretize subgoal space to improve the generalization ability of an HRL policy. The experimental results show the effectiveness of the proposed framework. However, since the paper is not well polished, it is hard to read and understand the keys under the hood. In addition, because there are many missing details in the experiment section, I have no confidence in the reproducibility. Finally, the usefulness of one of the claimed contributions about “expected policy gradients by A2C” is not validated, which makes the technical contributions weak.
> >
> > For your summary, we respectfully stress that our contribution is not only ‘propose a kind of generalizable subgoals’, but also ‘how to train the agent with the subgoal efficiently and stably and how to leverage these subgoals to generalize to unseen new tasks’. For the contribution of ‘expected policy gradient’, we show the result in the ‘Random-square Maze’. The environment is a changeable task with the stochastic initial position of the agent and the goal door. The corresponding generalization task is in an unseen room with an unseen structure. The adaptation of our agent in these complex tasks shows effectiveness. It is strong evidence because the tasks cannot be solved by previous SOTA methods or algorithms. For reproducibility, open-source code is helpful. We will upload our code to GitHub latter. Finally, we will check and polish our paper in the camera-ready version.
> >
> >
> >
> >
> > [1] Zhang, Huan, et al. "Robust deep reinforcement learning against adversarial perturbations on state observations." Advances in Neural Information Processing Systems 33 (2020): 21024-21037
> > [2] Fan, Linxi, et al. "SECANT: Self-Expert Cloning for Zero-Shot Generalization of Visual Policies." International Conference on Machine Learning. PMLR, 2021.
> > [3] Laskin, Misha, et al. "Reinforcement learning with augmented data." Advances in neural information processing systems 33 (2020): 19884-19895.
> > [4] Zhao, Chenyang, et al. "Investigating generalisation in continuous deep reinforcement learning." arXiv preprint arXiv:1902.07015 (2019).
> > [5] Goel, Vikash, Jameson Weng, and Pascal Poupart. "Unsupervised video object segmentation for deep reinforcement learning." Advances in neural information processing systems 31 (2018).

---

### Decision · Program_Chairs · 2023-01-20

**Decision:**

Reject

**Justification For Why Not Higher Score:**

There paper does not conduct experiments to accepted standards of practice nor does it provide evidence to how the design decisions enable better generalization. Overall this leads to possible misunderstandings in what is important for learning policies that generalize to multiple environments.

**Justification For Why Not Lower Score:**

N/A

**Metareview: Summary, Strengths And Weaknesses:**

This paper proposes a new combination of techniques for learning a hierarchical policy that will generalize across multiple environments. Specifically, using directional sub-goals over position-based sub-goals, adding noise into the sub-goals to force the high-level policy to learn a robust goal-setting strategy, and gradient averaging from multiple trajectories. The reviewers agree that directional sub-goals enable the agent to learn a more general policy. However, as the reviewers point out, there needs to be more evidence to understand how using randomness impacts the learning process or the agent's ability to generalize across environments. Furthermore, the experiments claim state-of-the-art performance but only use three trials and do not account for the hyperparameter selection process in the results. Addressing these gaps will make the results much more compelling and provide valuable insights into understanding the challenges of learning a policy that generalizes over environments.

**Summary Of Ac-Reviewer Meeting:**

N/A